# An Electrochemical Strategy for the Simultaneous Detection of Doxorubicin and Simvastatin for Their Potential Use in the Treatment of Cancer

**DOI:** 10.3390/bios11010015

**Published:** 2021-01-03

**Authors:** Iulia Rus, Mihaela Tertiș, Cristina Barbălată, Alina Porfire, Ioan Tomuță, Robert Săndulescu, Cecilia Cristea

**Affiliations:** 1Department of Analytical Chemistry, Iuliu Hațieganu University of Medicine and Pharmacy, 4 Louis Pasteur Street, 400349 Cluj-Napoca, Romania; Rus.Iulia@umfcluj.ro (I.R.); mihaela.tertis@umfcluj.ro (M.T.); rsandulescu@umfcluj.ro (R.S.); 2Department of Pharmaceutical Technology and Biopharmaceutics, Iuliu Hațieganu University of Medicine and Pharmacy, 41 Victor Babes Street, 400012 Cluj-Napoca, Romania; Barbalata.Cristina@umfcluj.ro (C.B.); aporfire@umfcluj.ro (A.P.); tomutaioan@umfcluj.ro (I.T.)

**Keywords:** doxorubicin, simvastatin, electrochemical simultaneous detection, pharmaceutical analysis

## Abstract

The aim of this study was to develop a disposable, simple, fast, and sensitive sensor for the simultaneous electrochemical detection of doxorubicin (DOX) and simvastatin (SMV), which could be used in preclinical studies for the development of new pharmaceutical formulations for drug delivery. Firstly, the electrochemical behavior of each molecule was analyzed regarding the influence of electrode material, electrolyte solution, and scan rate. After this, the proper electrode material, electrolyte solution, and scan rate for both active substances were chosen, and a linear sweep voltammetry procedure was optimized for simultaneous detection. Two chronoamperometry procedures were tested, one for the detection of DOX in the presence of SMV, and the other one for the detection of DOX and SMV together. Finally, calibration curves for DOX and SMV in the presence of each other were obtained using both electrochemical methods and the results were compared. The use of amperometry allowed for a better limit of detection (DOX: 0.1 μg/mL; SMV: 0.7 μg/mL) than the one obtained in voltammetry (1.5 μg/mL for both drugs). The limits of quantification using amperometry were 0.5 μg/mL for DOX (dynamic range: 0.5–65 μg/mL) and 2 μg/mL for SMV (dynamic range: 2–65 μg/mL), while using voltammetry 1 μg/mL was obtained for DOX (dynamic range: 1–100 μg/mL) and 5 μg/mL for SMV (dynamic range: 5–100 μg/mL). This detection strategy represents a promising tool for the analysis of new pharmaceutical formulations for targeted drug delivery containing both drugs, whose association was proven to bring benefits in the treatment of cancer.

## 1. Introduction

DOX, a compound isolated from natural sources, belongs to the anthracycline class and exhibits very potent activity against numerous types of cancer. Aside from its many pharmacological benefits in cancer therapy, DOX presents severe acute and long-term side effects, including multiple organ toxicity, developed in a dose-dependent manner [1,2]. In this regard, scientists explored many paths to reduce the toxicity related to DOX therapy, without altering its efficacy. One of the many proposed solutions was the association of DOX with statins.

Statins are widely used for their blood cholesterol lowering properties, but in the last years, more and more research has been carried out in regard to statins’ pleiotropic effects, i.e., antioxidant, anti-inflammatory, immune modulatory, and antiproliferative effects [3,4]. The antiproliferative effect of statins stems from their mechanism of action, namely the inhibition of 3-hydroxy-3-methyl-glutaryl-coenzyme A (HMGCoA) reductase in cholesterol synthesis [5]. Through this pathway, not only cholesterol synthesis is prevented, but also the synthesis of other intermediates, responsible for various processes in the development and metastasis of cancer [5,6]. The physico-chemical properties of statins are important for their antiproliferative effects, the majority of studies pointing out significantly higher antiproliferative activity for the lipophilic ones [7,8]. Among those, SMV is the most studied, with numerous in vivo and in vitro studies proving a synergistic or additive effect of SMV in combination with a large number of chemotherapeutic drugs, including DOX [9,10,11].

The benefits of using statins in cancer therapy were also confirmed through clinical trials, the most favorable outcomes being achieved in the case of colorectal cancer, prostate cancer, and breast cancer [12,13,14]. As the successful outcome of combinatorial anticancer therapy is often dependent on simultaneous co-exposure of tumor tissue to both agents, the development of targeted pharmaceutical systems able to co-deliver SMV and DOX in a spatial-temporal manner is currently of interest.

To date, liposomes are the most promising candidates meeting this goal, as they may incorporate both hydrophilic drugs, such as DOX hydrochloride, and lipophilic agents, such as SMV. In addition to the difficulties related to pharmaceutical development of such co-formulation, the analytical characterization is also challenging, a suitable method for the simultaneous assay of both active substances being needed. So far, two studies have evaluated the benefits of DOX and statins co-encapsulation in liposomes. In both cases, an inhibition of tumor growth due to synergistic effects between the two drugs was observed [15,16]. Regarding the characterization of liposomes, particularly entrapped drug concentration and drug release profile, it is worth mentioning that none of the studies managed to develop an analytical method for the simultaneous quantification of both drugs. In case of DOX, a spectrophotometric or fluorescent method was used, while for SMV or lovastatin, HPLC was the selected analytic strategy [15,16]. Additionally, neither of the studies provided a determination of the drug release profile on the co-formulation, excluding a possible influence of active substances on drug release processes, since there is a difference in their physicochemical properties. Given that the analytical strategy adopted by the two scientific groups is time and probe consuming, associated with a limitation in the type of analyzed samples, the development of a rapid analytical technique for simultaneous quantification of both active substances is a subject of scientific and practical importance.

As is well known, electrochemical analysis brings important advantages compared to classical analytical techniques (HPLC, spectrophotometry) such as simplicity, shorter analysis time, and reduced costs, offering at the same time comparable performance with HPLC regarding sensitivity, reproducibility, and limit of detection. Considering this and the benefits that the DOX and SMV combination brings in the treatment of cancer, a possibility of using electrochemistry for the analysis of new pharmaceutical formulations containing both drugs was taken into account.

This study describes for the first time the development of a simple and fast simultaneous detection of DOX and SMV using electrochemical techniques and screen printed disposable electrochemical cells.

## 2. Materials and Methods

DOX hydrochloride (>95% purity) bought from TCI, Tokyo, Japan, and SMV provided by Biocon Limited, Bangalore, India were used in this study. Screen printed electrodes (SPE) based on graphite, gold, and platinum, purchased from Metrohm DropSens, Oviedo, Spain, were used for the study of the influence of the electrode material. Pencil graphite electrode (PGE) with hard black (HB) hardness and 1 mm diameter bought from Rotring^®^ were also tested. Britton Robinson buffer (BRB), phosphate buffer (PB), phosphate buffer saline (PBS), acetate buffer (AB), and citrate buffer (CB) solutions were prepared with acetic, boric, phosphoric acid, sodium hydroxide, disodium hydrogen phosphate, potassium dihydrogen phosphate, citric acid, and sodium citrate from Sigma Aldrich, St. Louis, MI, USA. Ethanol (EtOH) from Sigma Aldrich and methanol (MeOH) from LiChroSolv, Merck, Darmstadt, Germany, of HPLC purity were used for the solubilization of SMV. In order to test the developed analytical techniques on real samples, two pharmaceutical formulations were used: Simvacard tablets containing 10 mg SMV produced by Zentiva^®^, Bucharest, Romania and Doxorubicin Accord^®^, Ahmedabad, India, 2 mg/mL, concentrated solution for infusion.

For the electroanalysis of DOX and SMV, two analytical techniques were used: linear sweep voltammetry (LSV) and chronoamperometry (CA). Many parameters that influenced the electrochemical behavior of the two molecules were analyzed, in order to choose the optimal detection conditions. The influence of the electrode material was first determined for DOX using SPE based on graphite (simple electrodes or modified with single-walled carbon nanotubes (SWCNTs)), gold nanoparticles (AuNPs modified electrodes) or platinum, and the electrodes with the best results were afterwards used for the detection of SMV. The influence of the solution’s pH was also evaluated, using 0.1 M AB of pH 3.23, 4.5, 5.53, CB of pH 6.77, PBS of pH 7.2 for DOX and BRB of pH 2–12 with 40% MeOH for SMV. The influence of scan rate variation on the analytical signal was also tested, using scan rates between 5–200 mV/s for DOX and 5–200 mV/s for SMV. An optimization of the ratio between the aqueous and alcoholic content of the SMV solution was performed before all tests.

Calibration curves were built for both substances, alone and in the presence of each other, using both analytical techniques. In voltammetry, solutions of 1–100 μg/mL DOX in the presence of 10 μg/mL SMV, as well as 20–100 μg/mL SMV in the presence of 10 μg/mL DOX were prepared and tested using 50 μL solution on graphite SPE. Solutions of the same concentration of DOX and SMV in the range of 5–150 μg/mL were prepared as well and tested by LSV.

In CA, calibration curves were built by adding specific volumes of 1 mg/mL DOX and SMV solutions, respectively, on 5 mL of PB pH 5 with 25% EtOH, under continuous stirring (600 rpm) after current stabilization and following the current intensity steps, obtaining calibration curves in the range of: 0.5–65 μg/mL for DOX alone, 2–65 μg/mL DOX in the presence of 10 μg/mL SMV, 2–65 μg/mL for SMV alone, and 2–45 μg/mL SMV in the presence of 10 μg/mL DOX. The intensity of the current for DOX was measured at 0.5 V, while for SMV at 0.95 V, these values being selected from cyclic voltammetry (CV) data performed in the same conditions. Finally, the concentrations obtained in the tested solution were calculated and the intensity of the current was graphically represented with the variation of the concentration.

Limits of detection (LODs) and limits of quantification (LOQs) were estimated in all cases. The LOQ was considered the lowest concentration tested, while the LOD was estimated based on the signal to noise ratio (S/N = 3.3).

For testing SMV from real samples, 10 tablets of 10 mg SMV were individually weighted and the average mass was calculated. After this, the tablets were triturated, and the average mass of a tablet was weighted and brought to 10 mL using EtOH. The suspension was sonicated for 1 h and then centrifuged for 10 min at 5000 rpm. From the clear resulting supernatant, dilutions of 50, 25, and 10 μg/mL were prepared in PB of pH 5 with 25% EtOH. From the DOX pharmaceutical form, dilutions of the same concentrations were prepared in PB of pH 5 with 25% EtOH. Furthermore, tests for the simultaneous detection of DOX and SMV were performed on solutions of equal concentration of the two drugs (10, 25, and 50 μg/mL) prepared from the individual pharmaceutical forms.

Selectivity studies were performed on the excipients of the pharmaceutical formulation (anhydrous lactose, pregelatinized starch, talcum, microcrystalline cellulose, butylhydroxyanisole, magnesium stearate, hydroxypropyl cellulose, hypromellose, and titanium dioxide for SMV; sodium chloride and hydrochloric acid for DOX).

The selectivity of the method was also evaluated in the presence of eventual phospholipids (1,2-dipalmitoyl-sn-glycero-3-phosphocholine; *N*-(carbonyl-methoxypolyethylenglycol-2000)-1,2-distearoylsn-glycero-3-phosphoethanolamine Na-salt) and cholesterol, as the main excipients used for the preparation of liposomes. For this purpose, 2 mL of liposomes (unloaded with active substances) containing 70 mM phospholipids were introduced in a Silde-A-Lyzer^®^ Dialysis Cassette (ThermoScientific, Waltham, MA, USA) and were kept under stirring in 100 mL of media (PB pH 5 with 25% EtOH) for 24 h. A simulated release medium containing phospholipids was obtained by performing the release study using blank liposomes (i.e., devoid of active substances), in the conditions in which the release study is performed for the active substances.

All the tests were performed using a multi-channel potentiostat/galvanostat Autolab MAC80100 (Metrohm, Utrecht, The Netherlands), operated with the specific Nova 1.10.4 software.

## 3. Results

### 3.1. The Electrochemical Characterization of Doxorubicin and Simvastatin

The main goal of this experimental study was to develop an easily operated, fast, and sensitive determination method, using an electro-analytical strategy without complicated preparations of the modified electrode for the simultaneous detection of DOX and SMV. For this purpose, the electrochemical behavior of each molecule was analyzed regarding the influence of electrode material, electrolyte solution, and scan rate. A LSV and an amperometric procedure were then optimized for the simultaneous detection of the two drugs using the proper electrode material, electrolyte solution, and scan rate.

#### 3.1.1. The Influence of the Electrode Material

The influence of the electrode material on the detection of DOX was evaluated on SPEs based on graphite, gold, and platinum. The highest intensity of the oxidation peak of DOX was obtained on graphite and SWCNTs modified electrodes. On 1 cm of isolated PGE of HB hardness and 1 mm diameter, good results were also obtained for DOX (Table 1).

Based on the results obtained in the case of DOX, only graphite-based electrodes were considered for the detection of SMV. Therefore, an oxidation peak was observed using graphite SPE, in voltammetry, at a potential around 1 V for SMV, while on PGE no analytical signal was observed in the potential range tested from −0.2 to +1.5 V (Table 1).

Considering the results presented in Table 1, for further tests, graphite disposable SPEs were chosen as the most suitable material for the simultaneous detection of DOX and SMV.

#### 3.1.2. The Influence of the Electrolyte and pH of the Solution

The redox process of DOX was followed on graphite SPE, using cyclic voltammetry (CV) in different buffer solutions of different pH as supporting electrolytes (Figure 1).

The results showed that when weak acidic and basic electrolytes were used, no oxidation signal for DOX was observed, suggesting that the oxidation process also involves the participation of protons. As can be observed in Figure 1, the intensity of the analytical current of DOX decreases with the increase of pH. If in AB solution of pH 3.23 (red) and pH 4.5 (green) the oxidation peaks have higher intensities, in the same buffer but of pH 5.53 (blue) the intensity decreases at about half the value obtained in more acidic media. In CB of pH 6.77 (orange) and PBS of pH 7.2 (magenta), the analytical signal of DOX was unnoticeable.

It was also noticed that increasing the pH value caused the oxidation peak of DOX to shift to lower potential values (Figure 2b; black), simultaneously with the decrease of the peak current intensity (Figure 2a; black), a phenomenon that may be explained either by the involvement of protons in the reaction that takes place at the electrode surface [17,18], either by the lower stability and solubility of DOX at more basic values of pH [19,20].

This type of behavior has been reported in other studies, so the proposed mechanism for the electrochemical oxidation of DOX involves an equal number of two protons and electrons, as it can be seen from the simplified mechanism presented in Figure 3. It was thus assumed that in the oxidation range of potential considered in this study, the well represented oxidation peak of DOX corresponds to the oxidation process of the hydroquinone center with the formation of a quinone, a process that involves two electrons and two protons [17,18,21].

As was described for the case of DOX, a higher intensity of the analytical signal of SMV was observed at lower values of pH, with the best results in the range of 5–6 (Figure 2a; red). Like in the case of DOX, no electrochemical oxidation of SMV was observed when basic electrolytes were used, leading to the presumption that the mechanism for SMV also involves the participation of protons. In the range of pH 2.0–12.0, the oxidation current increased with the decrease of pH at first and reached a maximum value at about pH 5. Further decrease of pH determined a slow decrease of the oxidation current, thus the effect of pH on the oxidation peak current presented a different type of variation compared with DOX. Regarding the influence of the pH on the oxidation potential of SMV, a slight increase of its value was observed with the variation of pH (Figure 2b; red). Thus, in the range of pH 2–12, the peak potential negatively shifted with the decrease of the pH value, without following a linear dependence on the whole pH range tested but only on small domains (Figure 2b; red). The abnormal variation of the peak potential with pH might arise from the coexistence of SMV and its hydrolyzed product since this analyte might be hydrolyzed at higher pH values, as it was already observed before in other papers [22].

Since for SMV the variation of the peak potential is smaller but still noticeable, it was assumed that an equal number of electrons and protons took part in the oxidation of SMV. (Figure 4). The chemical structure of SMV contains a *β*-hydroxy-lactone which undergoes a ring opening process during oxidation, as it has been previously observed [23].

Considering the low solubility of DOX at pH values higher than 5 and that the highest intensity of the analytical current of SMV was obtained at pH 5, further tests were performed in these conditions. Another reason for choosing pH 5 was that release studies of drug delivery systems with anticancer properties are often done at pH values lower than 6, being known that the pH at the tumor site is more acidic in comparison with healthy tissue [24,25]. Because phosphate buffers along with carbonate/bicarbonate buffers are the most used buffer solutions for in vitro release tests and because carbonate/bicarbonate buffer needs a permanent flow of CO_2_ in order to maintain the release conditions constant, PB was chosen as the proper media for being more similar to complex phosphate buffers [26,27].

#### 3.1.3. The Influence of the Scan Rate

The oxidation peak of DOX shifted towards more positive potential values, while the reduction peak slightly shifted towards more negative potential values (Figure 5A). This variation of the redox potential suggests that the electrochemical process is controlled by the adsorption of DOX molecules at the surface of the electrode but this assumption was not confirmed by the study of the correlation between the intensity of the analytical signal and scan rate (Figure 5B) and the precision of this correlation was poor. A similarly poor precision was observed by studying the correlation between the intensity of the analytical signal and square root of the scan rate as it can be observed in Figure 5C. Further tests were performed in order to clarify the mechanism of the electrochemical process and the plot of logI_ox_ versus logυ of DOX was represented. A linearity with the slope value of 0.345 was obtained (Figure 5D) this being close to the theoretical value 0.5, specific for diffusion-controlled processes [18,28], and far from 1, specific for adsorption-controlled processes. These results thus also suggest that the oxidation process of DOX is governed by diffusion of the analyte at the electrode’s surface, but in this case the precision of the correlation was also poor, therefore making it hard to draw very firm conclusion.

It was observed that on the graphite-based SPEs, the analytical signal of SMV, even though it is not so well defined, increased with the increase of the scan rate (Figure 6A). Studying the variation of the analytical signal with the scan rate (Figure 6B) and square root of the scan rate (Figure 6C), a better correlation with the scan rate was noticed, suggesting that the adsorption at the electrode’s surface is the process that governs the electrochemical oxidation of SMV (Inset in Figure 6). The plot of logI_ox_ versus logυ of SMV shows linearity, with the slope value of 1.04 (Figure 6D) which is close the theoretical value of 1, specific for adsorption-controlled processes. In all these cases the precision obtained was poor, but similar observations were reported in the literature for electrochemical oxidation of SMV on the glassy carbon electrode [23] and multi-walled carbon nanotubes-dihexadecyl hydrogen phosphate composite modified glassy carbon electrode [22], respectively.

### 3.2. The Optimization of the Simultaneous Detection of Doxorubicin and Simvastatin

#### 3.2.1. Simultaneous Detection of Doxorubicin and Simvastatin Using Voltammetry

Due to the significant difference between the oxidation potentials of DOX and SMV observed during the optimization of their individual detection, it was assumed that these analytes can be detected together using voltammetry. Because the scan rate study showed better results at moderate to high values of the scan rate, a LSV procedure with a scan rate of 100 mV/s was elaborated, optimized, and used for further studies.

The overlapped voltammograms obtained for different equal concentrations of DOX and SMV are presented in Figure 7. It can be observed that in the range of 5–150 μg/mL the intensity of the analytical signal is increasing with the increase in concentration for both drugs. Furthermore, the oxidation of DOX occurred at about 0.4 V while in the case of SMV it occurred around 1 V, meaning that the separation of the two peaks was approximately of 0.6 V, confirming the assumption of their simultaneous detection. However, for the SMV signal, a small peak situated at a lower value of potential appears like a shoulder attached to the main oxidation peak, without having a significant impact on the intensity of the analytical current (Figure 7). It was assumed that this pseudo-signal corresponds to the electrochemical oxidation process of SMV, which, under certain conditions of concentration and scan rate, takes place in two steps. This behavior was also observed when lower values of scan rate (<50 mV/s) and different ratios of aqueous and alcoholic fractions in the detection media were used.

Finally, calibration curves were built using this procedure for DOX in the presence of SMV, as well as for SMV in the presence of DOX (Figure 8).

Solutions of the same concentrations of DOX and SMV alone and in the presence of each other were prepared and tested in the same conditions. No significant difference was observed between DOX alone and DOX in the presence of SMV, as well as for SMV alone and SMV in the presence of DOX, these being observed from the calibration curves presented in Figure 8A,C, as well as from the overlapped voltammograms presented in Figure 8B,D.

For these results, the recovery of the analytical signal for DOX in the presence of SMV was 92.99%, and for SMV in the presence of DOX was 101.33%, compared with the analytical signal obtained for the same concentrations of each individual substance.

The figures of merit for the LSV simultaneous detection of DOX and SMV from solutions of equal concentrations of the two drugs using graphite based SPEs were determined. Thus, LOQs of 5 μg/mL, LODs of 1.6 μg/mL, and linear domains from 5 to 150 μg/mL were obtained.

#### 3.2.2. Simultaneous Detection of Doxorubicin and Simvastatin Using Amperometry

In order to eliminate the inconvenient presence of a shoulder on the main oxidation peaks of SMV observed in voltammetry and to obtain better LODs for both drugs, amperometry studies were performed. Therefore, the intensity of the current was followed using CA, at 0.95 V, in a buffer solution with 25% EtOH and successive additions of specific volumes of concentrated SMV solution (1 mg/mL) (Figure 9B). The same strategy was applied for DOX, in this case at a potential of 0.5 V (Figure 9A). By adding SMV between successive additions of DOX at a potential of 0.5 V, no variation in the intensity of the current was observed (Figure 9C). On the other hand, by adding DOX between successive additions of SMV at a potential of 0.95 V, an increase of the current was noticed (Figure 9D). In this particular situation, the signals registered for the oxidation of DOX, in solutions containing both substances, should be considered when calculating the amount of SMV.

The results obtained using LSV and amperometry for the simultaneous detection (different concentrations of DOX in the presence of fixed concentration of SMV: 0.01 mg/mL and vice versa) were compared with the ones obtained for mono-component solutions. The calibration curves for all situations are represented in Figure 10 and Table 2. It can be observed that when using both LSV and amperometry, the concentration ranges as well as the slope of the curves are very similar, proving the small influence of DOX and SMV for the detection of each other. Thus, analyzing the equations of the calibration curves obtained in all situations using all techniques (Table 2), a better LOD was observed using CA than LSV (LOD DOX: 0.3 μg/mL using LSV and 0.1 μg/mL using amperometry; LOD SMV: 1.5 μg/mL using LSV and 0.7 μg/mL using amperometry).

### 3.3. Selectivity Study and Real Samples Analysis

In order to test the analytical techniques developed in this study on real samples, tablets of SMV 10 mg and 2 mg/mL DOX concentrated solution for infusion were acquired and more diluted solutions (50, 25, and 10 μg/mL) were prepared from them to check the recovery of the drug concentration according to the one declared by the producer.

The optimized LSV detection strategy was applied for mono-component solutions and afterwards for solutions containing equal concentration of the two drugs prepared from pharmaceutical forms. The obtained recoveries are presented in Table 3.

The same solutions were also tested using the amperometry optimized detection strategies. The overlapping amperograms for the concentrations tested for both DOX and SMV are shown in Figure 11, compared to the signal recorded under the same experimental conditions in buffer (blank signal with black for DOX in Figure 11A and royal blue for SMV in Figure 11B). Thus, the increase of the current variation with the increase of the analyte concentration in the tested solution can be observed.

Recoveries were calculated using the equation of the calibration curves for DOX and SMV, using the optimized amperometric strategy.

In the case of SMV, the tablets contained the following excipients: anhydrous lactose, pregelatinized starch, talcum, microcrystalline cellulose, butylhydroxyanisole, magnesium stearate, hydroxypropyl cellulose, hypromellose, and titanium dioxide, which did not significantly interfere in the detection of SMV, after dissolution in EtOH and separation of the supernatant containing SMV via centrifugation. For this analyte, an average recovery of 97.00% (relative standard deviation (RSD) = 3.12%) was obtained for solutions of theoretical concentrations of 10, 25, and 50 μg/mL.

In the case of DOX, the concentrated solution for infusion contained sodium chloride and hydrochloric acid 0.1 M for the adjustment of the pH, none of them having an important influence on the detection of DOX. From the experimental results, an average recovery of 104.8% (RSD = 3.68%) was obtained for solutions of theoretical concentrations of 10, 25, and 50 μg/mL.

The simultaneous detection of DOX and SMV from pharmaceutical dosage forms containing both drugs could not be achieved in this study, due to the unavailability of this combination on the market. However, as mentioned in the introduction, liposomes are promising drug delivery systems to encapsulate both drugs, due to their capacity of encapsulating both hydrophilic and hydrophobic drugs. In the case of liposomes, a possible application of the developed method would be in the simultaneous assay of both active substances in the samples withdrawn from the release medium during the in vitro release studies.

Considering the mentioned application, a preliminary selectivity test was performed to highlight the influence of phospholipids and cholesterol traces, up to the maximum concentration in which they could be present in the release medium, on the detection of the targets. No signal was registered at both values of potential (0.5 V for DOX and 0.95 V for SMV) compared with clear medium, without phospholipids.

## 4. Conclusions

Various solutions have been searched for over time in order to improve the treatment of cancer. Combinatorial therapies along with drug delivery systems are promising tools for this purpose. An improvement of the efficacy of cancer therapy was observed when antitumor drugs were used together with statins.

In this study, new analytical strategies were described for the simultaneous detection of DOX and SMV, whose association is researched in several studies. Two electrochemical techniques, namely LSV and CA, were tested on graphite-based disposable SPEs, the last one proving to be more convenient, due to the better limit of detection obtained for SMV.

From the overall electrochemical results obtained in this study, it was concluded that the oxidation of DOX is reversible and it involves the exchange of two protons and two electrons, while for SMV it is irreversible and probably also involves an exchange of two protons and two electrons.

The interaction between DOX and SMV was also studied with excellent results, thus the optimized sensor seems to be promising and of great practical value for real life applications such as pharmacokinetic studies.

These analytical methods could be successfully used for the evaluation of the drug content and release profile of DOX and/or SMV from different pharmaceutical formulations in the preclinical stage of research.

## Figures and Tables

**Figure 1 biosensors-11-00015-f001:**
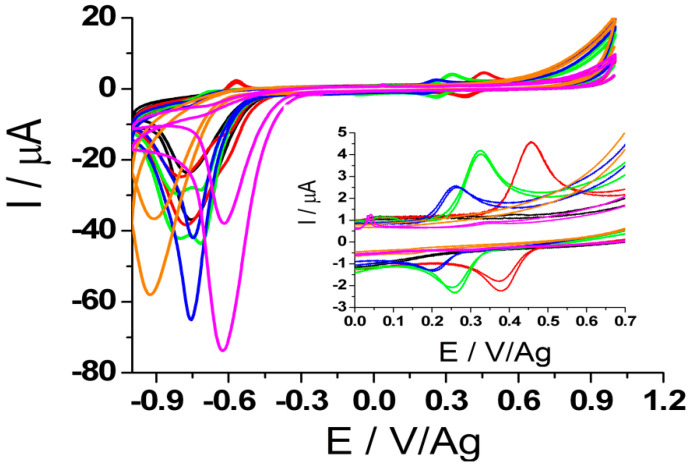
Cyclic voltammograms of 54.35 μg/mL (100 μM) DOX in: acetate buffer (AB) of pH 3.23 (red), 4.5 (green), 5.53 (blue), citrate buffer (CB) of pH 6.77 (orange), phosphate buffer saline (PBS) of pH 7.2 (magenta), recorded at SWCNTs modified graphite electrode (scan rate: 100 mV/s, potential range: (−1, +1 V)/Ag).

**Figure 2 biosensors-11-00015-f002:**
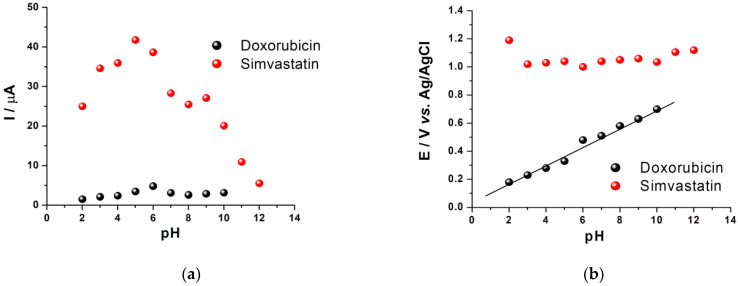
The influence of the pH on the intensity of the analytical signal (**a**) and oxidation potential (**b**) for 54.35 μg/mL (100 μM) DOX solution(black) and 1 mg/mL SMV solution (red), (scan rate: 100 mV/s, potential range: (−0.1, +0.8 V)/Ag for DOX and (−0.5, +1.6 V)/Ag for SMV.

**Figure 3 biosensors-11-00015-f003:**
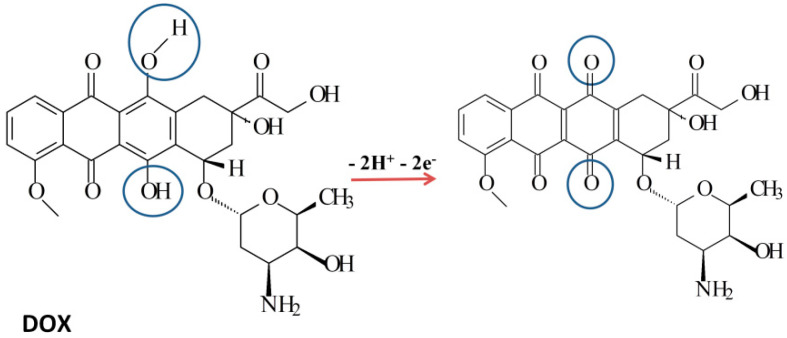
Proposed global mechanism for the electrochemical oxidation of DOX.

**Figure 4 biosensors-11-00015-f004:**
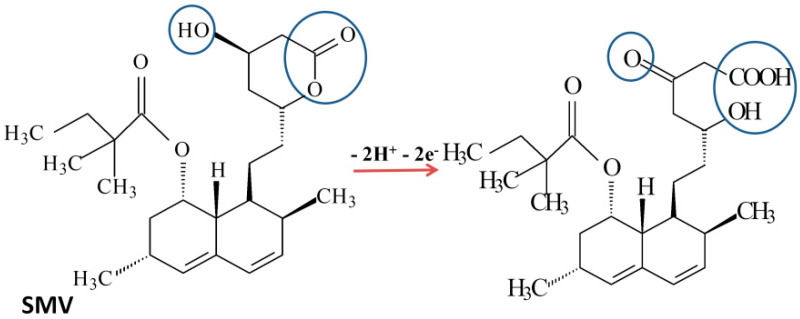
Proposed global mechanism for the electrochemical oxidation of SMV.

**Figure 5 biosensors-11-00015-f005:**
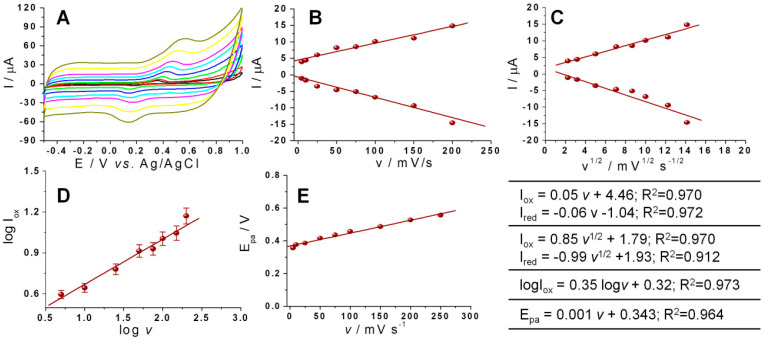
(**A**) Voltammograms registered for 100 μg/mL DOX solution in PB of pH 5 with 25% EtOH, using graphite SPE and different scan rates (5 mV/s: black; 10 mV/s: red; 25 mV/s: green; 50 mV/s: blue; 75 mV/s: light blue; 100 mV/s: pink; 150 mV/s: yellow and 200 mV/s: brown). The variation of the current peak intensity with the: (**B**) scan rate; (**C**) square root of the scan rate. (**D**) The variation of logI_ox_ with log*v*. (**E**) The variation of the oxidation potential with the scan rate. Inset in the right corner: equations of the above-mentioned correlations. Error bars were calculated based on the standard deviation obtained for 3 tests on 3 different electrodes.

**Figure 6 biosensors-11-00015-f006:**
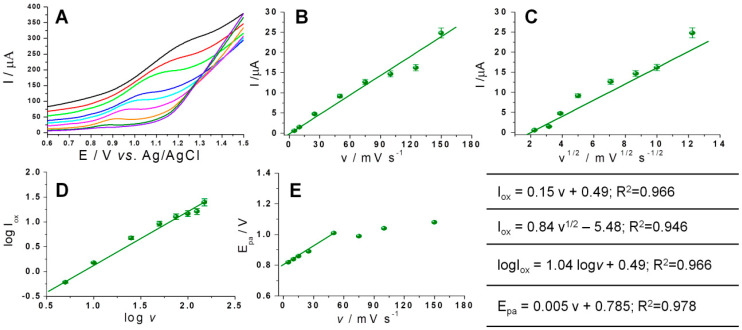
(**A**) Voltammograms registered for 100 μg/mL SMV solution in PB of pH 5 with 25% EtOH, using graphite SPEs and different scan rates (5 mv/s: purple; 10 mV/s: olive; 25 mV/s: orange; 50 mV/s: pink; 75 mV/s: light blue; 100 mV/s: blue; 150 mV/s: green; 175 mV/s: red; and200 mV/s: black). The variation of the current peak intensity with the: (**B**) scan rate; (**C**) square root of the scan rate. (**D**) The variation of logI_ox_ with log*v*. (**E**) The variation of the oxidation potential with the scan rate. Inset in the right corner: equations of the above-mentioned correlations. Error bars were calculated based on the standard deviation obtained for 3 testes on 3 different electrodes.

**Figure 7 biosensors-11-00015-f007:**
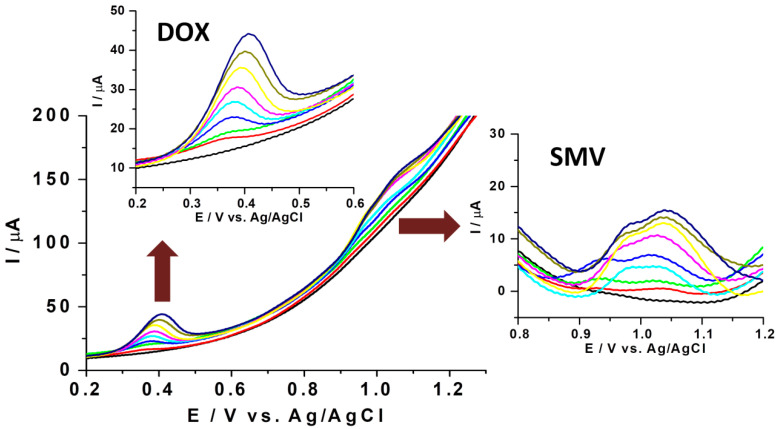
Voltammograms of different combinations of the same concentrations of DOX and SMV. Concentration range: 5–150 μg/mL (5 μg/mL: red; 10 μg/mL: green; 25 μg/mL: blue; 50 μg/mL: light blue; 75 μg/mL: pink; 100 μg/mL: yellow; 125 μg/mL: brown; and 150 μg/mL: royal blue; the signal registered in buffer in the absence of DOX and SMV: black); scan rate: 100 mV/s; potential range: (0, +1.5 V). Inserted figures: at the top—enlarged figure with the variation of the DOX signal with the concentration; at the right—enlarged figure with the variation of the SMV signal with the concentration.

**Figure 8 biosensors-11-00015-f008:**
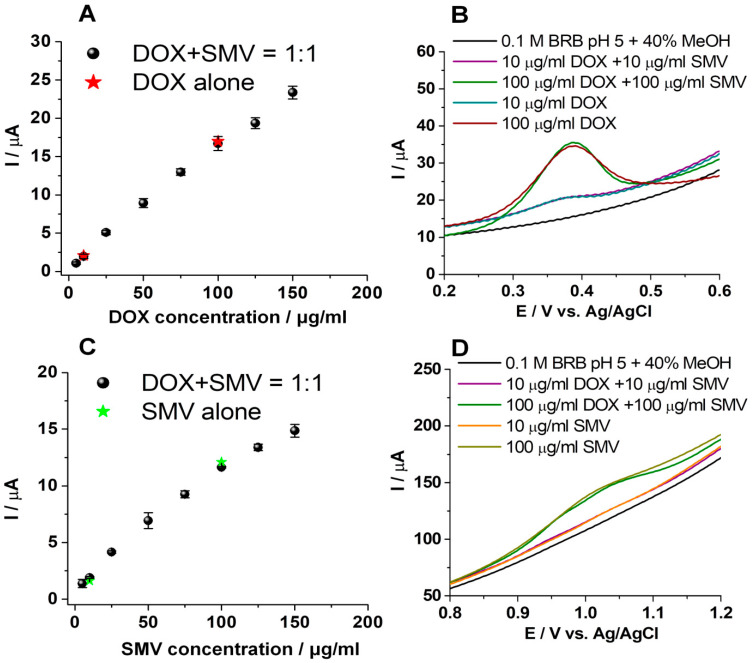
Calibration curves for DOX (**A**) and SMV (**C**) obtained by linear sweep voltammetry (LSV) in the presence of each other (equal concentrations of both analytes were tested). LSVs of different concentrations of DOX (**B**) and SMV (**D**) alone and in the presence of each other (see the legend of each figure for the correspondence between colors used and concentration). Scan rate: 100 mV/sl; potential range: (0, +1.5 V). Error bars were calculated based on the standard deviation obtained for 3 tests on 3 different electrodes.

**Figure 9 biosensors-11-00015-f009:**
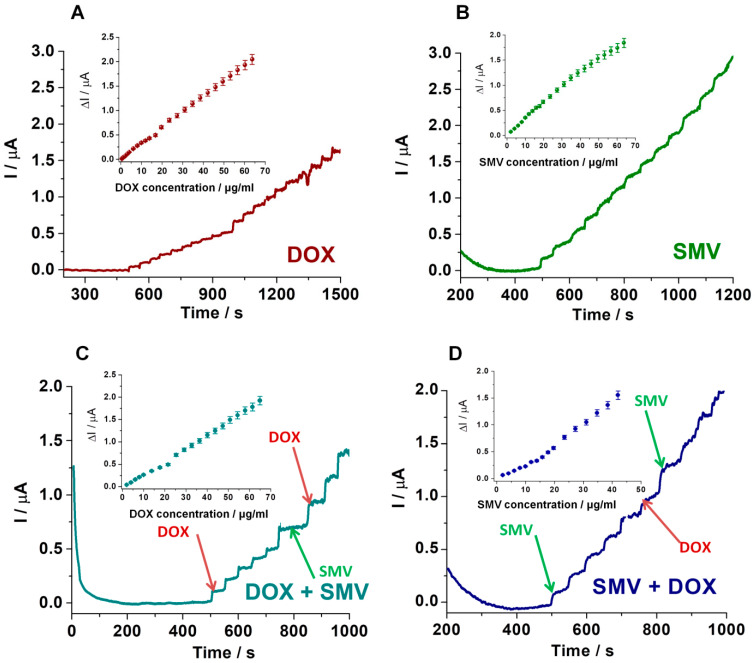
Chronoamperometry (CA) results obtained for successive additions of specific volumes of: 1 mg.mL DOX solution in 5 mL PB pH 5 + 25% EtOH at 0.5 V/Ag/AgCl (**A**); 1 mg.mL SMV solution in 5 mL PB pH 5 + 25% EtOH at 0.95 V/Ag/AgCl (**B**); **Inset:** calibrations curves of DOX and SMV alone using CA. Selectivity tests performed with amperometry for: addition of SMV between successive additions of DOX (**C**); addition of DOX between successive additions of SMV (**D**); **Inset:** calibrations curves of DOX in the presence of SMV and SMV in the presence of DOX, respectively (PB pH 5 + 25% EtOH, in CA at 0.95 V; under stirring conditions). Error bars were calculated based on the standard deviation obtained for 3 tests on 3 different electrodes.

**Figure 10 biosensors-11-00015-f010:**
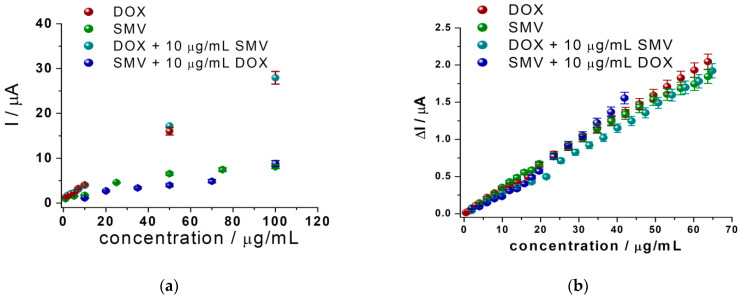
Calibration curves for DOX and SMV alone and in the presence of each other (DOX in the presence of constant concentration of SMV and vice versa) obtained using LSV (**a**) and amperometry (**b**) by using the optimized experimental conditions (see experimental conditions of Figure 8 for LSV and of Figure 9 for chronoamperometry). Error bars were calculated based on the standard deviation obtained for 3 tests on 3 different electrodes.

**Figure 11 biosensors-11-00015-f011:**
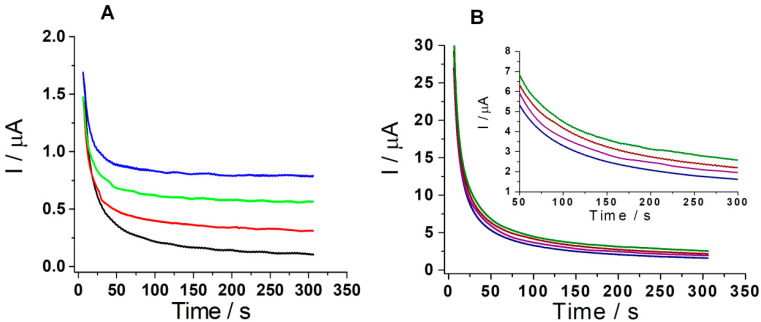
CAs for DOX (**A**) and SMV (**B**) quantification in solutions obtained from pharmaceutical forms (PB pH 5 + 25% EtOH, in CA at 0.5 V for DOX (blank test: black; 10 μg/mL—red; 25 μg/mL—green; 50 μg/mL—blue) and 0.95 V for SMV (blank test: royal blue; 10 μg/mL—purple; 25 μg/mL—wine; 50 μg/mL—olive); under stirring conditions, in the presence of excipients).

**Table 1 biosensors-11-00015-t001:** Results obtained with different electrode materials for the oxidation of doxorubicin (DOX) (54.35 μg/mL (100 μM); acetate buffer (AB) pH 3.23) and simvastatin (SMV) (50 μg/mL; Britton Robinson buffer (BRB) pH 5 + 40% MeOH).

Analyte	Electrode Material	E (V)	I (μA)
DOX	Graphite	0.428	2.935
	Graphite + AuNPs	0.431	2.935
	Gold	-	-
	Platinum	-	-
	PGE	0.525	10.818
SMV	Graphite	0.95	6.93
	PGE	-	-

**Table 2 biosensors-11-00015-t002:** Equations of the calibration curves for DOX and SMV alone and in the presence of each other, using LSV and Amperometry.

Analyte	LSV Data	Amperometry Data
DOX	I (μA) = 0.27 (DOX) (μg/mL) + 0.95 R^2^ = 0.999; Range: 1–100 μg/mL	I (μA) = 0.032 (DOX) (μg/mL) + 0.011 R^2^ = 0.999; Range: 0.5–65 μg/mL
DOX + 0.01 mg/mL SMV	I (μA) = 0.321 (DOX) (μg/mL) + 0.79 R^2^ = 0.986; Range: 1–100 μg/mL	I (μA) = 0.030 (DOX) (μg/mL) − 0.049 R^2^ = 0.997; Range: 2–65 μg/mL
SMV	I (μA) = 0.082 (SMV) (μg/mL) + 1.28 R^2^ = 0.996; Range: 5–100 μg/mL	I (μA) = 0.029 (SMV) (μg/mL) + 0.073 R^2^ = 0.994; Range: 2–65 μg/mL
SMV + 0.01 mg/mL DOX	I (μA) = 0.069 (SMV) (μg/mL) + 1.97 R^2^ = 0.914; Range: 20–100 μg/mL	I (μA) = 0.033 (SMV) (μg/mL) − 0.12 R^2^ = 0.987; Range: 2–45 μg/mL

**Table 3 biosensors-11-00015-t003:** Recoveries obtained for mono-component solutions of DOX and SMV and for solutions containing equal concentration of the two drugs prepared from pharmaceutical forms (10, 25, and 50 μg/mL; in BRB pH 5 + 25% EtOH).

Analyte	Expected Concentration (μg/mL)	Found Concentration (μg/mL)	Recovery (%)	RSD (%)
DOX	10	10.39	103.94	0.92
	25	28.98	115.93	4.11
	50	50.94	101.87	4.74
SMV	10	11.03	110.3	2.47
	25	25.79	103.19	6.08
	50	44.17	88.35	4.54
DOX(+SMV)	10	9.58	95.79	6.53
	25	24.70	98.80	0.26
	50	46.00	92.00	5.87
SMV(+DOX)	10	9.77	97.68	3.58
	25	24.70	98.80	5.73
	50	52.05	104.11	4.30

## Data Availability

The raw data corresponding to this study are available in the Department of Analytical Chemistry, Iuliu Hațieganu University of Medicine and Pharmacy and can be provided upon request by the corresponding author

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
