# Peer review of "An Electrochemical Strategy for the Simultaneous Detection of Doxorubicin and Simvastatin for Their Potential Use in the Treatment of Cancer"

_biosensors, 2021, doi:10.3390/bios11010015_

Round 1

Reviewer 1 Report

The authors developed a sensor for the detection of DOX and SMV but there are still some issues that need to be clarified before this manuscript can be accepted for publication.

The line numbers that I will mention are correspondent to the numbers on the side of the page.

  1. Page 1, line 18: Either active substances or medicines, not both.
  2. Page 1, line 20: At not in
  3. Page 2, line 50: were not was
  4. Page 2, line 62: Regarding instead of as regards
  5. Page 2, lines 63 to 65: Why is it necessary to develop a method to quantify DOX and SMV inside the liposomes? If liposomes are made in a solution with the desired ratio, why would this not be the ratio inside the liposomes?
  6. Page 2, lines 66 and 67: please correct the bookmark
  7. Page 2, lines 74 to 82: What I understood from these paragraphs was that the previous studies were performed in liposomes containing both DOX and a statin. Do you do the same here or the detection that you do in is not in liposomes? If is not in liposomes how can you compare the previous studies with yours? It is a very unfair comparison. If you are also doing the detection in liposomes loaded with both DOX and SMV you should make very clear here. (after reading the whole manuscript I understand that you did not use liposomes, but just the lipids)
  8. Page 3, line 106: When DOX is is loaded to the liposomes is it on a mixture of aqueous and alcoholic solutions? 
  9. Page 3, line 109: What are the Concentrations that are expected to be in the liposomes when applied to cancer treatment?
  10. Page 4, table 1: In what condition were these results obtained? Which pH?
  11. Page 4, Figure 1: The cyan curve is not seen, please change the colour.
  12. Page 5, line 176 to 181: How do you come up with this mechanism? The number of electrons being transferred can be determined from the CVs but how do you calculate the number of protons involved in the process?
  13. Page 5, lines 183 to 197: This paragraph must be re-written. There is some confusion on what the figures show. The authors confuse DOX with SMV and vice-versa.
  14. Page 6, lines 201 and 202: Let's assume that this is true (you should provide proper references), then how do you justify also the same number of electrons and protons being transferred for DOX? It presents a big variation in peak potential with the change in pH.
  15. Page 6, lines 206 to 209: Please re-write. Something is missing here.
  16. Page 6, lines 219 and 220: When and why does this happen?
  17. Page 6, lines 220 and 221: No! Diffusion. If it was adsorption the peaks would change only at very high scan rates and the peak separation should be very small. Also, Figure 5 does not show the characteristics of an adsorption process. Please lot log SR vs log i and check the slope. I'm guessing that it will be much closer to 0.5 than to 1. Also, to really prove adsorption you should perform a wider range of scan rates.
  18. Page 6, line 225: 0.345 is far from 0.5 but is even further from 1, which would correspond to adsorption.
  19. Page 7, table 2 and page 8, table 3: these tables are useless. t would be more relevant it the authors would present the log SR vs log i plots.
  20. Page 9 figure 7: Why don't you merge 7a and 7b? It would save space to present the log figures mentioned above. Same goes for 7c and 7d.
  21. Page 9 figure 7: From the data presented in table 1, the current generated by the 54.35 ug/mL of DOX is 2.935 uA, which means (assuming linear relationship) that 100 ug/mL should generate 5.4 uA. That 5.4 uA are comparable to the 4.545 generated by 100 ug/mL of SMV (value from table 1). Why here we see a huge difference in the current generated by equal amounts of DOX and SMV? (for 100 ug/mL DOX generates about 30 uA and SMV less than 10 uA).
    Also, why are the values in table 1 so different from the ones presented here?
  22. Page 10, figure 8: what is B? Also, B does not show anything. Note that the current for 0 concentration, between 400 and 500 s) is the equal to the current after 4 additions of DOX, so basically you cannot detect DOX in this range.
  23. Page 11, figure 9a: This figure does not show anything! Actually is contradicts the data shown in figure 7b.
  24. Page 13, graphical abstract: This is not a very good graph to show here because what it shows is that you cannot detect SMV, while you claim to detect it.

Author Response

Answer for Reviewer 1

Response: We thank the reviewer for carefully reading our manuscript and for the observations that allow us to increase its quality. We have answered to all questions and we hope that our response will clarify all the issues highlighted by the reviewer.

The authors developed a sensor for the detection of DOX and SMV but there are still some issues that need to be clarified before this manuscript can be accepted for publication.

The line numbers that I will mention are correspondent to the numbers on the side of the page.

Response: The manuscript was carefully read and expression, typos and English errors were corrected. We hope that the revised form meets the requirements of the journal.

  1. Page 1, line 18: Either active substances or medicines, not both.

Response: Correction was done.

  1. Page 1, line 20: At not in

Response: Correction was done.

  1. Page 2, line 50: were not was

Response: Correction was done.

  1. Page 2, line 62: Regarding instead of as regards

Response: Correction was done.

Page 2, lines 63 to 65: Why is it necessary to develop a method to quantify DOX and SMV inside the liposomes? If liposomes are made in a solution with the desired ratio, why would this not be the ratio inside the liposomes?

Response:When preparing liposomes, the active substances are not 100% loaded inside the liposomal vesicles. The preparation procedure implies a purification step during which the unloaded active substances are eliminated. For this reason, it is necessary to quantify DOX and SMV inside the liposomes, to determine the concentration of each in the liposomal dispersion and the encapsulation efficiency (the concentration of the incorporated active substance over the initial concentration used to prepare the formulation).

  1. Page 2, lines 66 and 67: please correct the bookmark

Response: Correction was done.

  1. Page 2, lines 74 to 82: What I understood from these paragraphs was that the previous studies were performed in liposomes containing both DOX and a statin. Do you do the same here or the detection that you do in is not in liposomes? If is not in liposomes how can you compare the previous studies with yours? It is a very unfair comparison. If you are also doing the detection in liposomes loaded with both DOX and SMV you should make very clear here. (after reading the whole manuscript I understand that you did not use liposomes, but just the lipids)

Response: The purpose of the study was to develop a new detection strategy for DOX and SMV that could be used to evaluate the capacity of loading and in vitro release of various drug delivery systems. Simultaneous detection of these two drugs is primarily envisaged since there are recent studies on their synergistic action when used together in cancer treatment. Liposomes were mentioned as the optimal drug delivery system due to their ability to encapsulate both hydrophilic and lipophilic drugs and the previous studies presented were meant to highlight the necessity of a new analytical method for the simultaneous detection of the two drugs. This study describes the steps followed in the development of the method, including interferences studies with phospholipids (the main compounds of liposomes), but no determination of DOX and SMV from liposomes were performed to date.

  1. Page 3, line 106: When DOX is is loaded to the liposomes is it on a mixture of aqueous and alcoholic solutions? 

Response: Liposomes are lipid vesicles dispersed in an aqueous phase. To determine the concentration of active substances in liposomes, the vesicles are dissolved in ethanol or another solvent in which both the lipids and the active substance are soluble. This is the reason why the solvent used for the calibration curves was a mixture of aqueous and alcoholic solutions.

  1. Page 3, line 109: What are the Concentrations that are expected to be in the liposomes when applied to cancer treatment?

Response: The concentrations used in cancer treatment depend on the sensitivity of the tumor cells to the toxicity of the combination. However, when the concentrations are determined in liposomes, the liposomal dispersion is diluted 100 to 200 times with mixtures of ethanol and water/PB. The calibration range was set as to cover the concentrations found in the liposomes after a dilution of the specified order and also to cover the concentrations found in the release medium, during in vitro release studies.

For example, Doxil is the first liposomal form of DOX, approved by FDA in 1995 (https://www.accessdata.fda.gov/drugsatfda_docs/label/2007/050718s029lbl.pdf), and it has a concentration of 2 mg/mL DOX hydrochloride in liposome injection (concentrated solution for infusion). Therefore, the limit of detection of the developed analytical method for DOX is adequate, a dilution of the release samples being even necessary. In the case of SMV, there are still not available drug delivery systems, but the encapsulation of SMV together with DOX was discussed in many studies. In a study by Li et al, the loading content obtained for SMV was 1.81%, while for DOX was 1% (see reference [15] in the manuscript).

  1. Page 4, table 1: In what condition were these results obtained? Which pH?

Response: The data regarding the influence of the electrode material on the oxidation of DOX and SMV presented in Table 1 were revised in the case of SMV (data corresponding to a concentration of 50 μg/ mL, closer to that used for DOX, were now included instead of those obtained for 100 μg/ mL). For the study of the influence of the electrode material on the detection of DOX, a solution of 54.35 μg/mL DOX (equivalent to 100 μM DOX) in acetate buffer solution of pH 3.23 was used, while for SMV, a solution of 50 μg/mL SMV in Britton Robinson buffer solution of pH 5 with 40% MeOH was needed due to the low solubility of SMV in aqueous media.

The data regarding the media electrolytes used in this part of the study were included in the description of Table 1.

  1. Page 4, Figure 1: The cyan curve is not seen, please change the colour.

Response: The quality of Figure 1 was improved as the reviewer suggested.

  1. Page 5, line 176 to 181: How do you come up with this mechanism? The number of electrons being transferred can be determined from the CVs but how do you calculate the number of protons involved in the process?

Response: The mechanisms for the oxidation of DOX and SMV has been proposed by comparing the experimental data obtained within this study with other studies found in the literature. Thus, in the studies cited in the manuscript as references [17], [18], [21] and [23] similar behavior regarding the shift of the peak potentials to lower values with the increasing of pH is attributed to a process that involves equal number of electrons and protons.

  1. Page 5, lines 183 to 197: This paragraph must be re-written. There is some confusion on what the figures show. The authors confuse DOX with SMV and vice-versa.

Response: This paragraph has been revised as suggested by the reviewer and we hope that the message sent is clearer now. The corrections made are marked with track changes.

  1. Page 6, lines 201 and 202: Let's assume that this is true (you should provide proper references), then how do you justify also the same number of electrons and protons being transferred for DOX? It presents a big variation in peak potential with the change in pH.

Response: The mechanism for the oxidation of DOX and SMV has been proposed by comparing the experimental data obtained within this study with other studies in the literature. Thus, in the studies cited in the manuscript as references [17], [18], [21] and [23] a similar behavior regarding the shift of the peak potential to lower values with the increasing of pH is attributed to a process that involves equal number of electrons and protons.

In the revised version of the manuscript, all the experimental data regarding the electrochemical behavior of DOX and SMV (eg. the study of the variation of the peak current/ peak potential with pH, the study of the variation of the peak current/ peak potential with scan rate, the variation of the logarithm of the peak current with the logarithm of scan rate, etc.) were included to prove the assumptions made regarding the mechanisms of the oxidation processes of the two drugs at the electrode (see Figure 2, Figure 5 and Figure 6).

  1. Page 6, lines 206 to 209: Please re-write. Something is missing here.

Response: This paragraph was revised and we hope that in this form it is clearer! All the changes were marked with track changes.

  1. Page 6, lines 219 and 220: When and why does this happen?

Response: This paragraph was revised and we hope that in this form it is clearer. Furthermore, more experimental data were included to support the presented discussions. All the changes were marked with track changes.

  1. Page 6, lines 220 and 221: No! Diffusion. If it was adsorption the peaks would change only at very high scan rates and the peak separation should be very small. Also, Figure 5 does not show the characteristics of an adsorption process. Please lot log SR vs log i and check the slope. I'm guessing that it will be much closer to 0.5 than to 1. Also, to really prove adsorption you should perform a wider range of scan rates.

Response: The revised version of the manuscript shows the variation of the logI with logv for both DOX and SMV, as suggested by the reviewer. On the scan rate range tested here, the type of mechanism that seems dominant for each analyte was suggested, and the results are discussed in the text with reference to the corresponding data in literature. All the changes were marked with track changes in the manuscript.

  1. Page 6, line 225: 0.345 is far from 0.5 but is even further from 1, which would correspond to adsorption.

Response: The corresponding graphs were included in Figure 5 for DOX and Figure 6 for SMV, respectively. Discussions for these new figures have also been provided and we hope that in this form they are clearer and consistent with the data presented. All the changes were marked with track changes in the revised manuscript.

  1. Page 7, table 2 and page 8, table 3: these tables are useless. t would be more relevant it the authors would present the log SR vs log i plots.

Response: Tables 2 and 3 have been removed as suggested by the reviewer. The data presented in these tables have been included in the revised versions of Figure 5 for DOX and Figure 6 for SMV. These figures now contain several grouped graphs:

  1. the CVs (in the case of DOX) and LSVs (in the case of SMV) showing the variation of the current on the potential range tested when applying different scan rates, in the range from 5 to 200 V/s;
  2. Variation of the intensity of the peak current with the scan rate
  3. Variation of the intensity of the peak current with the square root of the scan rate
  4. Variation of the logarithm of the intensity with the logarithm of the scan rate;
  5. Variation of the peak potential with the scan rate.

The equations that characterize all these correlations between different parameters and scan rate are also presented in these two figures: in Figure 5 for DOX, and in Figure 6 for SMV, respectively (see the revised version of the manuscript and the revised version of all the figures).

  1. Page 9 figure 7: Why don't you merge 7a and 7b? It would save space to present the log figures mentioned above. Same goes for 7c and 7d.

Response: Figure 7 became Figure 8 in the revised manuscript and the graphs were merged as suggested by reviewer. In the same time, calibration curves were represented for DOX in the presence of equal concentration of SMV and vice versa to demonstrate the simultaneous detection of these two compounds using the LSV optimized procedure proposed in the manuscript.

  1. Page 9 figure 7: From the data presented in table 1, the current generated by the 54.35 ug/mL of DOX is 2.935 uA, which means (assuming linear relationship) that 100 ug/mL should generate 5.4 uA. That 5.4 uA are comparable to the 4.545 generated by 100 ug/mL of SMV (value from table 1). Why here we see a huge difference in the current generated by equal amounts of DOX and SMV? (for 100 ug/mL DOX generates about 30 uA and SMV less than 10 uA).
    Also, why are the values in table 1 so different from the ones presented here?

Response: First of all, in Table 1, the results presented for DOX are obtained using a solution of DOX in a different media than the one used for the data presented in Figure 7. Data presented in Table 1 were obtained for a solution of DOX prepared in acetate buffer of pH 3.23, while the ones presented in Figure 7 were obtained for a solution of DOX in BRB of pH 5, with 40% MeOH. The influence of the media can explain de differences between the results obtained. In the case of SMV, the current intensity was revised (data corresponding to a concentration of 50 μg/ ml, closer to that used for DOX were now included instead of those obtained for 100 μg/ ml).

  1. Page 10, figure 8: what is B? Also, B does not show anything. Note that the current for 0 concentration, between 400 and 500 s) is the equal to the current after 4 additions of DOX, so basically you cannot detect DOX in this range.

Response: Figure 8 became Figure 9 in the revised version of the manuscript and 2 more graphs were added: CAs obtained for successive additions of specific volumes of: 1mg/mL DOX in 5 mL PB pH 5+25% EtOH at 0.5 V/ Ag/AgCl and 1mg/mL SMV solution in 5 mL PB pH 5+25% EtOH at 0.95 V/ Ag/AgCl, respectively. The other 2 graphs represent additions of SMV between successive additions of DOX at 0.5 V and additions of DOX between successive additions of SMV at 0.95 V. Thus, it can be observed that by adding SMV in the analysed solution at 0.5 V, the intensity of the current doesn’t suffer any change, the oxidation of SMV having no place at this value of potential. On the other hand, by adding DOX between successive additions of SMV at 0.95 V, the oxidation of DOX takes place, which means that using this procedure, both molecules will be detected. Therefore, using chronoamperometry, SMV can be quantified in the presence of DOX by making the difference between the current intensity obtained at 0.95 V and the current intensity obtained at 0.5 V.

  1. Page 11, figure 9a: This figure does not show anything! Actually is contradicts the data shown in figure 7b.

Response: We thank the reviewer for this observation. Figure 9a became Figure 10a in the revised manuscript. Indeed, for the detection of SMV alone with LSV the concentration range was different from the one used for SMV in the presence of constant amount of DOX. By reducing the concentration range for SMV alone to the one for SMV in the presence of constant amount of DOX, it can be observed that the slope of the curves have closer values suggesting that there is no big influence of the DOX presence for the LSV detection of SMV. The same thing was also observed for the influence of SMV on the LSV detection of DOX (see Figure 10(a))

  1. Page 13, graphical abstract: This is not a very good graph to show here because what it shows is that you cannot detect SMV, while you claim to detect it.

Response: Graphical abstract was revised. The simultaneous detection of the two drugs is represented in the revised version of the manuscript via LSV, being thus suggestive for the main subject of the study.

Reviewer 2 Report

this manuscript describe an electrochemical method for detection of Dox and SMV. It is a interesting study however several issues need to be address before the considering for publicaiton:

  1. the title says "novel" simutaneous detection". actually this electrochmical  method is not novel. Both of Dox and SMV are electroactive specicials. thus they can be detected by electrochemistry. In addition, the actual method introduced here can not be used for simutaneous detection based on the current setup, and they need to be detected seperatly by fixing at specific potentia at 0.5 v or 0.9 V, to my understanding. thus the title needs to be revised.
  2. the importance of simutaneous detection of Dox and SMV is well justified in the introduction. but the useful concentration of Dox and SMV needs to be specified, which is importance to define the target sensitivity of the method. Is the senstivity of this method for Dox and SMV clinical relevant or useful for the real application?
  3. In addition, the potential for SMV is very high, and at such high potentil, there will be lots of interferences which can happen electrochemcial reaction. how to eliminate the interferences needs to be clarified.
  4. For a sensor, the sensing interface is very important, hoe to make sure the sensing interface is reproducilable and reliable?
  5. For SMV redox peaks between 0.9 and 1.1 v, it looks like two peaks there, how to expalin it and how to quantify the current responding to SMV?
  6. There are lots of English errors and typos, such as in the same time (should be at the same time), in the same condition (should be under the same conditions), etc

Author Response

Answer for Reviewer 2

We thank the reviewer for carefully reading our manuscript and for the observations that allow us to increase its quality.

This manuscript describe an electrochemical method for detection of Dox and SMV. It is a interesting study however several issues need to be address before the considering for publicaiton:

  1. the title says "novel" simutaneous detection". actually this electrochmical  method is not novel. Both of Dox and SMV are electroactive specicials. thus they can be detected by electrochemistry. In addition, the actual method introduced here can not be used for simutaneous detection based on the current setup, and they need to be detected seperatly by fixing at specific potentia at 0.5 v or 0.9 V, to my understanding. thus the title needs to be revised.

Response: The main detection strategy from this study was based on a LSV optimized method which allows the simultaneous detection of both DOX and SMV in an experiment which takes about 15 seconds. New data, hopefully more suggestive, regarding this simultaneous detection were added in the revised version of the manuscript (Figure 7, and Figure 8). The amperometric detection strategies for DOX and SMV were applied as an alternative in the case of samples having lower content of analytes than the LOD of the LSV technique (new figures with amperometric data were also included: Figure 9, Figure 10, Figure 11; all the other figures were revised for increase quality). The novelty of the study states in the description for the first time of the simultaneous detection of the two drugs using electrochemistry.

  1. the importance of simutaneous detection of Dox and SMV is well justified in the introduction. but the useful concentration of Dox and SMV needs to be specified, which is importance to define the target sensitivity of the method. Is the senstivity of this method for Dox and SMV clinical relevant or useful for the real application?

Response: The main purpose of this study was to develop an analytical method suitable for the simultaneous detection of SMV and DOX in loading/relese studies, for which the LODs obtained are suitable. For example, Doxil which is the first liposomal form of DOX, approved by FDA in 1995 (https://www.accessdata.fda.gov/drugsatfda_docs/label/2007/050718s029lbl.pdf), and it has a concentration of 2 mg/mL DOX hydrochloride in liposome injection (concentrated solution for infusion). Therefore, the limit of detection of the developed analytical method for DOX is adequate, a dilution of the release samples being even necessary. In the case of SMV, there are still not available drug delivery systems, but the encapsulation of SMV together with DOX was discussed in many studies. In a study by Li et al, the loading content obtained for SMV was 1.81%, while for DOX was 1% (see reference [15] in the manuscript).

The sensitivity of the method is important especially for the determination of DOX and SMV in the release media, during the in vitro relase studies, as the concentration of the active substances, especially at the beginning of the study, may go down to 0,1% of the concentration of the 2 substances in the liposomes.

Moreover, the LODs obtained in this study allowed the quantification of SMV and DOX from individual pharmaceutical forms, this consisting in another aplicability of the analytical method.

  1. In addition, the potential for SMV is very high, and at such high potentil, there will be lots of interferences which can happen electrochemcial reaction. how to eliminate the interferences needs to be clarified.

Response: In order to eliminate the interferences a sample of the clear media will always be analyzed in the same conditions, and the result will be considered by making the difference between the current intensity of the sample and the current intensity of the blank. Moreover, having developed both the LSV and the amperometric detection, the testing of samples by both methods can be useful for mutual control and thus the presence of some interferences can be discovered more easily.

  1. For a sensor, the sensing interface is very important, hoe to make sure the sensing interface is reproducilable and reliable?

Response: In the present study, commercial electrochemical screen-printed cells for which there are guarantees from the manufacturer regarding the reproducibility and reliability of the working surface were used. Also, comparative studies were performed which resulted in very good values regarding the accuracy of the signal recorded on several electrodes tested under the same conditions in the presence of a redox probe. Reproducibility problems are eliminated if they occur by tests performed periodicaly in buffer or in solutions with a known concentration of the redox probe, and the elimination of those with significant differences.

  1. For SMV redox peaks between 0.9 and 1.1 v, it looks like two peaks there, how to expalin it and how to quantify the current responding to SMV?

Response: Indeed, for SMV signal, a small peak situated at a lower value of potential, appears like a shoulder attached to the main oxidation peak, without having a significant impact on the intensity of the analytical current (see Figure 7). It was assumed that this pseudo-signal corresponds to the electrochemical oxidation process of SMV, which, under certain conditions of concentration and scan rate, takes place in two steps. This behaviour was also observed when lower values of scan rate (<50 mV/s) and different ratios of aqueous and alcoholic fractions in the detection media were used.

  1. There are lots of English errors and typos, such as in the same time (should be at the same time), in the same condition (should be under the same conditions), etc

Response: The manuscript was carefully read and language, typos and English errors were corrected. We hope that the revised form meets the requirements of the journal.

Reviewer 3 Report

The presented manuscript describes an electrochemical method for the simultaneous determination of two drugs: doxorubicin and simvastatin on unmodified  or modified screen printed electrodes. In my opinion, this work should not be published in Biosensors. The electrodes presented in the paper below cannot be called biosensors, because they do not meet this definition. In my opinion, publishing this work in Sensors MDPI can be considered after major revision.

My comments and questions:

  1. Overall, the text is quite hard to read. Often the work lacks typical analytical phrases, for example a range instead of linear dynamic range or concentration range. This makes it difficult to find important information in the manuscript. Therefore, I must say that the manuscript is sloppy.
  2. Abstract should be more consise and contain specific information, such as LODs, LOQs and LDRs.
  3. The quality of the proposed mechanisms of DOX and SMV oxidation are really poor and not acceptable to publish.
  4. Figures 3, 5 and 6 should contain trend lines with equations and R2 coefficients describing them.
  5. 6 – Why the correlation of the scan rate with the square root is not typical straight line? Could authors explain that?
  6. In my opinion, authors should prepare table contains results for the electrochemical determination of DOX and SMV by other scientist and discuss that.
  7. The correctness of the abbreviations should be checked (plural forms), for example SWCNTs and AuNPs instead of SWCNT and AuNP.
  8. Did authors checked the maximum residue limits for this drugs? Please do that and discuss that values with obtained LODs and LOQs.
  9. The voltammograms and amperograms for concentration dependance of studied drugs should be showed in the manuscript.

Author Response

Answer for Reviewer 3

We thank the reviewer for carefully reading our manuscript and for the observations that allow us to increase its quality.

We have answered to all questions and we hope that our response will clarify all the issues highlighted by the reviewer.

The presented manuscript describes an electrochemical method for the simultaneous determination of two drugs: doxorubicin and simvastatin on unmodified  or modified screen printed electrodes. In my opinion, this work should not be published in Biosensors. The electrodes presented in the paper below cannot be called biosensors, because they do not meet this definition. In my opinion, publishing this work in Sensors MDPI can be considered after major revision.

My comments and questions:

  1. Overall, the text is quite hard to read. Often the work lacks typical analytical phrases, for example a range instead of linear dynamic range or concentration range. This makes it difficult to find important information in the manuscript. Therefore, I must say that the manuscript is sloppy.

Response: The manuscript was carefully read and expression, typos and English errors were corrected. We hope that the revised form meets the requirements of the journal.

  1. Abstract should be more consise and contain specific information, such as LODs, LOQs and LDRs.

Response: The suggested information was added in abstract.

  1. The quality of the proposed mechanisms of DOX and SMV oxidation are really poor and not acceptable to publish.

Response: The quality of the mechanism was improved.

  1. Figures 3, 5 and 6 should contain trend lines with equations and Rcoefficients describing them.

Response: All figures presented in the manuscript have been revised for better quality or containing additional information. Also, new figures have been inserted being accompanied by appropriate explanations and discussions (see the revision version of the manuscript in which all the modification were made with track changes).

  1. Why the correlation of the scan rate with the square root is not typical straight line? Could authors explain that?

Response: The dependence between the current intensity and the square root of the scan rate is not always linear, but only in the case of electrochemical transformation processes governed by the diffusion of electrochemically active species at the electrode to perform electrons transfer. In the case of processes governed by other phenomena, such as adsorption/ desorption of species on the surface, this dependence is not linear. In the range of scan rates tested in this study, it was observed that for DOX we have diffusion as the rate-determining step, while for SMV oxidation the adsorption is the rate-determining step (see Figure 5 and Figure 6 and the corresponding explanations in the manuscript). Our opinion, confirmed by some studies in the literature, is that it is most likely a competition between the two phenomena, diffusion and adsorption, both having a certain contribution in the global mechanism of electrochemical transformation of DOX and SMV.

  1. In my opinion, authors should prepare table contains results for the electrochemical determination of DOX and SMV by other scientist and discuss that.

Response: To the best of our knowledge, this study is the first one which describes the use of LSV and CA for the simultaneous detection of DOX and SMV.

Thus, we consider that comparison between our study and other studies in literature presenting only the individual detection of DOX or SMV would not be relevant.

  1. The correctness of the abbreviations should be checked (plural forms), for example SWCNTs and AuNPs instead of SWCNT and AuNP.

Response: The manuscript has been completely revised; all abbreviations in the text have been revised and are now in the correct form.

  1. Did authors checked the maximum residue limits for this drugs? Please do that and discuss that values with obtained LODs and LOQs.

Response: The main purpose of this study was to develop an analytical method suitable for the simultaneous detection of SMV and DOX in loading/relese studies, for which the LODs obtained are suitable. For example, Doxil which is the first liposomal form of DOX, approved by FDA in 1995 (https://www.accessdata.fda.gov/drugsatfda_docs/label/2007/050718s029lbl.pdf), and it has a concentration of 2 mg/mL DOX hydrochloride in liposome injection (concentrated solution for infusion). Therefore, the limit of detection of the developed analytical method for DOX is adequate, a dilution of the release samples being even necessary. In the case of SMV, there are still not available drug delivery systems, but the encapsulation of SMV together with DOX was discussed in many studies. In a study by Li et al, the loading content obtained for SMV was 1.81%, while for DOX was 1% (see reference [15] in the manuscript).

Moreover, the LODs obtained in this study allowed the quantification of SMV and DOX from individual pharmaceutical forms, this consisting in another aplicability of the analytical method.

  1. The voltammograms and amperograms for concentration dependance of studied drugs should be showed in the manuscript.

Response: The voltammograms for different concentrations of DOX and SMV were inserted in the revised manuscript as new Figure 7, while the amperograms requested by the reviewer were presented in the new Figure 9. We hope that all the modifications and data inserted in the manuscript as figures or discussions/explanations have made the manuscript clearer and easier to understand.

Round 2

Reviewer 1 Report

I still do not agree with the way that the authors assume the oxidation mechanisms of DOX and SMV, so I my opinion is reject the article. I think more experiments (and other techniques) are needed, and the authors have not provided that in this review.

Author Response

Answer for Reviewer 1

We thank the reviewer for carefully reading our revised manuscript and answer.

Reviewer 1

I still do not agree with the way that the authors assume the oxidation mechanisms of DOX and SMV, so I my opinion is reject the article. I think more experiments (and other techniques) are needed, and the authors have not provided that in this review.

Response: Regarding this observation, we want to highlight the fact that both mechanisms of electrochemical oxidation of DOX and SMV were proposed based on the experimental results we obtained using CV and LSV, respectively, which were compared with the available data from literature.

Thus, in a study by Deepa et al, cited in our article as reference [18] (doi:10.1016/j.jelechem.2020.114748), the same oxidation mechanism involving two electrons and two protons was suggested for DOX based on similar voltammetry data obtained using carbon paste electrode. In another recent study (https://doi.org/10.1016/j.sintl.2020.100033), the same mechanism was proposed, this time on pencile graphite electrode. The diffusion controlled electrochemical oxidation of DOX was emphasized in both studies. As it can be observed from the CV’s presented in our study, the electrochemical oxidation of DOX on graphite SPE is a reversible process, other studies describing the reduction of DOX being also available in the literature. In this article, all the observations we made were focused on the oxidation process, thus, the comparison was done only with studies in which oxidation was approached.

Regarding the proposed mechanism for SMV electrochemical oxidation, the results presented in this study showed an irrevesible process, involving two protons and two electrons, as it was also suggested in other studies (Journal of Chemical and Pharmaceutical Research, 2012, 4(5):2803-2816, doi:10.3390/molecules24122215).  In the majority of studies from the literature, the electrochemical transformation of SMV is controlled by adsorbtion in competition with diffusion, as it was also observed in our study.

The electrochemical oxidative degradation of DOX and SMV should be performed in order to obtain the redox pathways for both drugs. Due to the inaccessibility to MS/MS technology which is absolutely necessary for this purpose, we could not add more data to support the proposed mecanisms.

Reviewer 2 Report

The authors have addressed the inquiries by adding new figures and revisions. it is suggested to be considered for publication.

Author Response

Answer for Reviewer 2

The authors have addressed the inquiries by adding new figures and revisions. it is suggested to be considered for publication.

We thank the reviewer for carefully reading our revised manuscript and answer and for the positive feedback.

Reviewer 3 Report

In my opinion, publishing this work in Bioensors MDPI can be considered after minor revision.

My comments and questions:

  1. Abstract should be more consise and contain specific information, such as LODs, LOQs and LDRs. These values are not contained in abstract till now. 

Author Response

Answer for Reviewer 3

We thank the reviewer for carefully reading our revised manuscript and answer and for the positive feedback.

Reviewer 3

In my opinion, publishing this work in Bioensors MDPI can be considered after minor revision.

My comments and questions:

Abstract should be more consise and contain specific information, such as LODs, LOQs and LDRs. These values are not contained in abstract till now.

Response: The abstract section of the manuscript was revised as suggested and all the data required were inserted with track changes (page 1, lines 25-29). 
